# The Effect of S-15176 Difumarate Salt on Ultrastructure and Functions of Liver Mitochondria of C57BL/6 Mice with Streptozotocin/High-Fat Diet-Induced Type 2 Diabetes

**DOI:** 10.3390/biology9100309

**Published:** 2020-09-24

**Authors:** Natalia V. Belosludtseva, Vlada S. Starinets, Lyubov L. Pavlik, Irina B. Mikheeva, Mikhail V. Dubinin, Konstantin N. Belosludtsev

**Affiliations:** 1Laboratory of Mitochondrial Transport, Institute of Theoretical and Experimental Biophysics, Russian Academy of Sciences, Institutskaya 3, Pushchino, 142290 Moscow, Russia; vlastar@list.ru (V.S.S.); pavlikl@mail.ru (L.L.P.); mikheirina@yandex.ru (I.B.M.); bekonik@gmail.com (K.N.B.); 2Department of Biochemistry, Cell Biology and Microbiology, Mari State University, pl. Lenina 1, Yoshkar-Ola, 424001 Mari El, Russia; dubinin1989@gmail.com

**Keywords:** type 2 diabetes mellitus, S-15176 difumarate salt, mitochondrial dysfunction, mitochondrial biogenesis, mitochondrial dynamics

## Abstract

**Simple Summary:**

Type II diabetes mellitus (T2DM) is one of the most common diseases, which currently represents a major medical and social problem due to the chronic course, high rates of disability and mortality among patients. Mitochondria of the liver and other vital organs are one of the main targets of T2DM at the intracellular level. The pathological changes in the structure of mitochondria, hyperproduction of reactive oxygen species by the organelles, disorders in mitochondrial transport systems and ATP synthesis are now widely recognized as important factors in the development of diabetes. Therefore, treatment strategies to attenuate mitochondrial injury may result in cellular reprogramming and alleviation of the diabetes-related pathological complications. The aim of present work was to investigate the possible protective effect of S-15176, a potent derivative of the anti-ischemic agent trimetazidine, against mitochondrial damage in the liver of mice with experimental T2DM. The data indicate that S-15176 attenuates mitochondrial dysfunction and ultrastructural abnormalities in the liver of T2DM mice. The mechanisms underlying the protective effect of S-15176 are related to the stimulation of mitochondrial biogenesis and the inhibition of lipid peroxidation in the organelles. One may assume that the compound acts as a mitochondria-targeted metabolic reprogramming agent in T2DM.

**Abstract:**

S-15176, a potent derivative of the anti-ischemic agent trimetazidine, was reported to have multiple effects on the metabolism of mitochondria. In the present work, the effect of S-15176 (1.5 mg/kg/day i.p.) on the ultrastructure and functions of liver mitochondria of C57BL/6 mice with type 2 diabetes mellitus (T2DM) induced by a high-fat diet combined with a low-dose streptozotocin injection was examined. An electron microscopy study showed that T2DM induced mitochondrial swelling and a reduction in the number of liver mitochondria. The number of mtDNA copies in the liver in T2DM decreased. The expression of *Drp1* slightly increased, and that of *Mfn2* and *Opa1* somewhat decreased. The treatment of diabetic animals with S-15176 prevented the mitochondrial swelling, normalized the average mitochondrial size, and significantly decreased the content of the key marker of lipid peroxidation malondialdehyde in liver mitochondria. In S-15176-treated T2DM mice, a two-fold increase in the expression of the PGC-1α and a slight decrease in *Drp 1* expression in the liver were observed. The respiratory control ratio, the level of mtDNA, and the number of liver mitochondria of S-15176-treated diabetic mice tended to restore. S-15176 did not affect the decrease in expression of *Parkin* and *Opa1* in the liver of diabetic animals, but slightly suppressed the expression of these proteins in the control. The modulatory effect of S-15176 on dysfunction of liver mitochondria in T2DM can be related to the stimulation of mitochondrial biogenesis and the inhibition of lipid peroxidation in the organelles.

## 1. Introduction

Type II diabetes mellitus (T2DM) is one of the most common diseases, which currently represents a major medical and social problem due to the chronic course, high rates of disability, and mortality among patients associated with diabetes-related complications. The pathogenesis of T2DM involves the development of insulin resistance, which leads to an increased level of blood glucose (hyperglycemia) and metabolic disorders. Chronic hyperglycemia contributes to a suppression of insulin secretion and can cause pathological changes in the structure and functions of many organs and systems of the organism, including the development of nephropathy, cardiovascular diseases, retinopathy, and other severe pathologies [1,2].

Mitochondria of the liver and other vital organs are known to be one of the main targets of T2DM at the intracellular level [3,4]. It was shown that the disorders in the system of oxidative phosphorylation, hyperproduction of reactive oxygen species (ROS), alterations in the mitochondrial transport systems responsible for calcium homeostasis, and remodeling of the mitochondrial network can contribute to the development of insulin resistance and T2DM [5]. The mitochondrial dysfunction of diabetic animals is coupled to the impairment of mitochondrial biogenesis, the dysregulation of the master metabolic regulator of mitochondrial biogenesis and respiratory function PGC-1α, and the occurrence of ultrastructural defects [3,4,5,6]. Therefore, treatment strategies to attenuate mitochondrial injury in the main body tissues may result in cellular reprogramming and alleviation of the diabetes-related pathological complications.

As known, a number of antidiabetic drugs are able to improve the mitochondrial function [7,8,9,10]. Mitochondria-targeted agents are also found to exhibit antidiabetic effect to varying degrees of effectiveness [11,12,13]. An increase in the β-oxidation of fatty acid in mitochondria is known to stimulate gluconeogenesis, which, in its turn, is an important pathogenic factor during the development of diabetic hyperglycemia [14]. The liver has a large capacity for mitochondrial β-oxidation of fatty acids, which is critical for systemic metabolic adaptation. So, the limitation of the fatty acid oxidation in liver mitochondria by the inhibition of carnitine palmitoyltransferase I (CPT-1) may be one of the mechanisms to control hyperglycemia and insulin resistance. Indeed, specific inhibitors of CPT-1 were observed to have a pronounced hypoglycemic effect [15,16,17]. Another therapeutic strategy for the management of diabetes and obesity may be to use the mitochondrial uncoupling agents, which can contribute to both the suppression of increased ROS production by mitochondria and the loss of body weight [11,12,18]. The specific blockers of the mitochondrial permeability transition (MPT) pore were also tested as potential antidiabetic compounds [13,19].

In this work, we have examined the effects of S-15176 difumarate salt ((N-[(3,5-Di-tert-butyl-4-hydroxy-1-thiophenyl)]-3-propyl-N′-(2,3,4-trimethoxybenzyl)piperazine difumarate salt) on ultrastructural and functional alterations of mitochondria of the liver of mice with experimental T2DM. The drug is a derivative of the effective anti-ischemic agent trimetazidine and possesses a broad spectrum of action. A possible therapeutic effect of S-15176 can be related with its ability to inhibit CPT-1, to cause mild mitochondrial uncoupling, to decrease the excessive production of ROS (antioxidant effect), and to suppress the opening of the MPT pore in mitochondria [20,21,22,23,24]. Taking together, these effects of S-15176 difumarate salt may prevent mitochondrial injury occurring in T2DM.

In view of the above, the aim of the present work was to investigate the effects of S-15176 (1.5 mg/kg/day i.p. for 20 days) on the development of mitochondrial dysfunction in the liver of mice with T2DM. The following results were obtained: (1) S-15176 at a dose applied had no hypoglycemic effect, but partially reversed ultrastructural abnormalities in the liver of T2DM mice; (2) upon the development of T2DM, liver mitochondria swell and become hypertrophied; the treatment of diabetic mice with S-15176 preserved ultrastructure and normalized the average size of the organelles; (3) S-15176 treatment led to a two-fold increase in the expression level of *PGC-1α* (peroxisome proliferator-activated receptor gamma coactivator 1-α) and a normalization of *Drp1* (dynamin-1-like protein) expression; (4) T2DM led to an enhancement of lipid peroxidation in mouse liver mitochondria; the administration of S-15176 to diabetic mice returned this parameter to the control value; (5) the administration of S-15176 did not affect the decrease in expression levels of *Opa1* and *Parkin* in the liver of diabetic animals, but led to a slight decrease in those of control mice; (6) S-15176-treated diabetic mice presented a tendency to restore the respiratory control ratio, the number of mtDNA copies, and the total number of liver mitochondria. The results of the study indicate that S-15176 partially suppresses the development of mitochondrial dysfunction in the liver of T2DM mice. Together, our data provide insight into the mechanisms underlying the action of S-15176 on liver mitochondria in healthy and diabetic animals.

## 2. Materials and Methods

### 2.1. Animals

Four- to five-week-old male C57BL/6 mice weighing 14 to 16 g were used. The animals were purchased from the Animal Breeding Facility, Branch of the Shemyakin and Ovchinnikov Institute of Bioorganic Chemistry, Russian Academy of Sciences, (IBCh RAS Unique Research Device “Bio-model”, Pushchino, Russia). All manipulations with animals were performed in accordance with the European Convention for the Protection of Vertebrates used for experimental and other purposes (Strasbourg, 1986) and the principles of the Helsinki Declaration (2000). All the protocols were approved by the Ethics Committees at the the Institute of Theoretical and Experimental Biophysics, Russian Academy of Sciences (Protocol No. 13/2020 of 17.02.2020).

### 2.2. Induction of Diabetes

Male C57BL/6 mice were randomly divided into four groups: (1) non-treated control (CTR) (*n* = 5); (2) control + S-15176 (CTR + S-15176) (*n* = 5); (3) T2DM (*n* = 6); and (4) mice with T2DM treated with S-15176 (T2DM + S-15176) (*n* = 6). T2DM was induced in mice from the third and the fourth groups by high-fat diet feeding (Adjusted Calories Diet 60/Fat (ACD), Envigo, Indianapolis, INUSA, #TD.06414) for four weeks [25]. This was followed by daily administration of a low dose of streptozotocin (STZ) intraperitoneally (i.p.) (35 mg/kg body weight, freshly made before the injection) dissolved in ice-cold 0.1M citrate buffer (pH 4.5) for five consecutive days in continuation with ACD feeding. On the 33rd day, ACD was withdrawn, and the mice were maintained on a normal balanced diet within four weeks. The control groups 1 and 2 were provided with low-fat control diet (Envigo, Indianapolis, IN, USA, #TD.08806) and drinking water ad libitum, followed by vehicle, i.e., 0.1M citrate buffer (pH 4.5) given i.p. On the 40th day, mice from groups 2 and 4 were treated with S-15176 (1.5 mg/kg/day) i.p. for 20 days. The dose selection was based on the available data on the drug effect on the activity of CPT-1. At the dose used, S-15176 inhibited the activity of the mitochondrial enzyme in liver tissue [20]. S-15176 was dissolved in DMSO/saline buffer. The body weight (BW) was measured weekly for each animal with an accuracy within 0.1 g. The successful induction of T2DM was confirmed by the intraperitoneal glucose tolerance test (IPGTT) and intraperitoneal insulin sensitivity test (IPIST).

### 2.3. Intraperitoneal Glucose Tolerance Test and Intraperitoneal Insulin Sensitivity Test

Before starting the tests, mice were fasted overnight (8:00 p.m. to 09:00 a.m.) with free access to water [26]. The level of blood glucose was measured at time zero, before a solution of glucose (2 g/kg of BW) in 0.1 mL of distilled water was administered i.p. to perform the IPGTT [24,25]. Afterwards, the blood glucose level was monitored by tail bleeding at 15, 30, 60, and 120 min after glucose injection using a One Touch Select Plus glucometer (LifeScan Inc., Zug, Switzerland). To perform the IPIST, human insulin was administrated at a dose of 1U/kg i.p., after which the levels of blood glucose were measured at 15, 30, 60, and 120 min. For the IPGTT and IPIST, the area under the curve (AUC) in the range from 0 to 120 min after challenge was calculated to quantitatively evaluate glucose clearance activity [27].

### 2.4. Electron Microscopy

For the electron microscopy study, samples of the liver were taken from the edge of the left lateral lobe and fixed for 2 h in a 2.5% glutaraldehyde solution in 0.1 M phosphate-buffered saline (PBS, pH 7.4). After washing with the buffer, the tissue was fixed for 2 h with a 1% solution of osmium acid in PBS and dehydrated by increasing concentrations of alcohols. The resulting samples were encapsulated in Epon 812 resin. Ultrathin sections (70–75 nm) were prepared on a Leica EM UC6 microtome (Leica Microsystems, Wetzlar, Germany) and stained with uranyl acetate and lead citrate. The preparations were examined and photographed using a JEM-100B electron microscope (JEOL Ltd., Tokyo, Japan). Ultrastructural analysis was performed using negative images digitized with an Epson V700 scanner. The morphometric analysis of images was conducted on photographic negatives using the Image Tool software.

### 2.5. RNA Extraction, Reverse Transcription, and Quantitative Real-Time PCR

Total RNA was isolated from 100 mg of deep-frozen tissue samples of the liver (the edge of the right lateral lobe) using an ExtractRNA kit (Cat. No. BC032, Eurogen, Moscow, Russia) in accordance with the protocol of the manufacturer. The real-time PCR was performed on a DTLite5 amplifier (DNA-Technology LLC, Moscow, Russia) using the qPCRmix-HS SYBR reaction mixture (Eurogen, Moscow, Russia). The selection and analysis of gene-specific primers were performed using Primer-BLAST [28] (the oligonucleotide sequences are presented below; Table 1). The relative level of expression of each gene was normalized to the level of Rplp2 mRNA, and a comparative C_T_ method was used to quantify the results [29].

### 2.6. Quantification of Mitochondrial DNA

Total DNA (nuclear and mtDNA) was extracted from 10 mg of the mouse liver (the edge of the right lateral lobe) using a DNA-Extran 2 kit (Cat. No. EX-511, Sintol, Russia) in accordance with the protocol of the manufacturer. One nanogram of the total DNA was taken for the reaction. The mtDNA content in the liver tissue was evaluated by PCR, as described [30] and expressed as mtDNA/nuclear DNA ratio. For the assay, we selected the *ND4* gene of the mitochondrial genome and the *GAPDH* gene of the nuclear one. A comparison of *ND4* DNA expression relative to *GAPDH* DNA expression will give a measure of mtDNA copy number to nDNA copy number ratio. Primers for mtDNA and nDNA are presented in Table 1. The real-time PCR was performed on a DTLite5 amplifier (DNA-Technology LLC, Moscow, Russia) using the qPCRmix-HS SYBR reaction mixture (Eurogen, Moscow, Russia), which contained the commonly used fluorescent DNA binding dye SYBR Green II.

### 2.7. Isolation of Liver Mitochondria

Mitochondria were isolated from liver tissue by differential centrifugation, as described earlier [31]. The homogenization buffer contained 210 mM mannitol, 70 mM sucrose, 1 mM EDTA, and 10 mM Hepes/KOH buffer, pH 7.4. Subsequent centrifugations were performed in the same buffer, except that, instead of EDTA, 100 μM EGTA was used. Final suspensions contained 40 to 50 mg of mitochondrial protein/mL, as determined by the Lowry method [32].

### 2.8. Mitochondrial Respiration and Oxidative Phosphorylation

The rate of oxygen consumption by isolated mitochondria was measured polarographically with a Clark-type gold electrode Oxygraph-2k (O2k, OROBOROS Instruments, Innsbruck, Austria) at 25 °C under continuous stirring [31]. The reaction medium contained 130 mM KCl, 5 mM KH_2_PO_4_, 2.5 mM potassium malate, 2.5 mM potassium glutamate 10 µM EGTA, and 10 mM Hepes/KOH (pH 7.4). The concentration of mitochondrial protein in the cuvette was 0.3–0.5 mg/mL. The following reagents were added: 200 μM ADP, 50 μM 2,4-dinitrophenol (DNP). Energetic state definitions: state 2, basal substrate respiration; state 3, respiration stimulated by addition of ADP; state 4, the mitochondrial state after all ADP is depleted; state 3U_DNP_, respiration in the presence of the uncoupling agent DNP. The respiratory control ratio (RCR) was calculated as the ratio of respiration rates in state 3/state 4. The rates of oxygen consumption by mitochondria were expressed as nmol O_2_ × min^−1^ × mg^−1^ of protein.

### 2.9. Lipid Peroxidation

Lipid peroxidation in a suspension of isolated mitochondria was estimated spectrophotometrically by measuring the levels of thiobarbituric acid-reactive substances (TBARS). The TBARS assay quantifies the levels of malondialdehyde and other minor aldehyde species through their reaction with thiobarbituric acid. The concentration of TBARS was calculated using the molar absorption coefficient of the colored TBA–MDA complex (E_535_ = 1.56 × 10^5^ M^−1^ × cm^−1^) [33].

### 2.10. Statistical Analysis

The data were analyzed using the GraphPad Prism 7.0 (GraphPad Software, San Diego, CA, USA) and were presented as mean ± SEM of 3 to 6 experiments. The statistical significance of differences between the experimental groups was evaluated using one-way analysis of variance (ANOVA) followed by the Tukey multiple comparison post-hoc test or the Kruskal-Wallis test after testing a normality and equal variance in the data sets. The differences were considered statistically significant at *p* < 0.05.

## 3. Results

### 3.1. Somatic and Biochemical Characteristics of Mice

Table 2 shows the data on the change in body weights of C57BL/6 mice of four groups over the experimental two-month period. One can see that the weight gain of animals from the T2DM group was higher compared to the control. Diabetic mice treated with S-15176 gained almost the same weight as the animals in the control group.

Figure 1A shows blood glucose levels during the IPGTT of C57BL/6 mice in four experimental groups measured at 0, 15, 30, 60, and 120 min after glucose injection. In control groups, blood glucose reached a maximum level after 15 min of glucose administration. In diabetic animals, the level of blood glucose increased to a maximum level after 30 min of glucose injection. It should be noted that the increase in the glucose level was significantly higher in diabetic mice compared to control animals. So, there was a delay of glucose clearance in diabetic mice, with glucose levels remaining elevated for 120 min after glucose administration, which permits one to characterize this state as glucose intolerance. The blood glucose levels of S-15176-treated mice with T2DM remain almost unchanged compared to that of untreated diabetic animals. This indicates that S-15176 had no hypoglycemic effect. 

The IPIST (Figure 1B) showed a rapid decline in plasma glucose after 15 min and a subsequent increase by 60 min of insulin administration in both control and diabetic mice. At the same time, blood glucose levels remained higher in diabetic mice than in control ones at all time points up to 120 min, indicating the development of insulin resistance in diabetic mice. The administration of S-15176 to both control and diabetic animals did not affect the shape of blood glucose curves. Thus, S-15176 did not affect insulin sensitivity. 

The data of IPGTT and IPIST were also analyzed in terms of the total AUC between 0 and 120 min (Figure 1C,D, respectively). In both tests, the AUC value was significantly higher in diabetic mice as compared to control animals. At the same time, there were no statistically significant differences between the AUC values in S-15176-treated mice and those in mice without S-15176 injection.

### 3.2. The Effect of S-15176 Difumarate Salt on Ultrastructural Changes in Liver Mitochondria of Mice with Experimental T2DM

Recent reports implicate that S-15176 difumarate salt exerts a number of effects on mitochondria. As shown by in vivo and in vitro studies, the drug can suppress the activity of mitochondrial CPT-1, the opening of the MPT pore, and overproduction of ROS by mitochondria [19,20,21,22,23]. So, the next objective of our work was to examine how this compound would affect the structural alterations in liver mitochondria of mice with T2DM.

Figure 2 shows the ultrastructural features of liver mitochondria of mice from the experimental groups. In the control, mitochondria are oval or elongated, possess well-packed cristae and an electron-dense matrix without disorders in crista position, and have a size typical for the liver (Figure 2A). It can be seen that rough endoplasmic reticulum (rER) is well developed and studded with numerous ribosomes. Free ribosomes and polysomes are detected between the membrane-enclosed compartments of rER. Single primary lysosomes and small lipid inclusions are also observed. One can see that mouse liver mitochondria in experimental groups 1 (control) and 2 (control + S-15176) are of much the same structure (Figure 2B). Mitochondria appear as polymorphic structures (from round to elongated in the shape) with an electron-dense matrix. 

The induction of T2DM in mice (group 3) results in morphological changes in hepatic mitochondria: the organelles become swollen and hypertrophied (Figure 2C,E). Some mitochondria show the vacuolation of cristae and disintegration accompanied by the loss of components of the mitochondrial matrix (Figure 2E). In addition, a decrease in the amount of the rER cisterns with a simultaneous proliferation of smooth endoplasmic reticulum (sER) is observed. One can see that the number of primary lysosomes increases and, in the area of their accumulation, pronounced destructive changes in mitochondria are detected. 

A distinguishing feature of the S-15176 + T2DM group is the prevalence of the round-shaped mitochondria with electron dense matrix and well-packed cristae (Figure 2D,F). At the same time, some mitochondria are small in size. As in group 3, both a decrease in the amount of rER cisterns and an increase in that of sER cisterns in the cytoplasm are observed. In addition, most mitochondria form close topographic connections with rER cisterns (Figure 2D).

Figure 3A shows a histogram of size distribution of mitochondria in each experimental group. It can be seen that the perimeter of mitochondria isolated from the liver of control animals is predominantly equal to ~1.6 μm (45% of all mitochondria). In control animals treated with S-15176, the perimeter of liver mitochondria slightly (statistically insignificant) increases to 1.6–2.4 μm (30% and 37% of all examined mitochondria, respectively). In the course of the development of diabetes, the main mitochondrial population is divided into three size groups: 1.6, 2.4, and 3.2 μm (21–23% of mitochondria of a particular size in each group). Besides, 5% of all mitochondria in the T2DM group are less than 1 μm in size (micro-mitochondria). As seen in Figure 3B, the development of T2DM lead to a decrease in the number of liver mitochondria (the number of mitochondria per plate). It should be noted that the differences in the number and the size of liver mitochondria between the CTR and T2DM groups are statistically significant.

In S-15176-treated diabetic mice, the perimeter of 40% of all liver mitochondria is ~1.6 μm, i.e., is close to that in the control. Meanwhile, a population of micro-mitochondria (8%) in this experimental group was found. Further analysis of the data indicates that there are statistically significant differences in the size distribution of mitochondria between the T2DM and the T2DM+S-15176 groups. The size distribution of liver mitochondria from the mice of the CTR and T2DM+S-15176 groups does not statistically differ. This suggests that treatment with S-15176 leads to a normalization of the average size of liver mitochondria of mice with T2DM. Moreover, S-15176-treated mice showed a tendency to restore the number of liver mitochondria, but the difference in the parameter value between the T2DM and T2DM+S15176 groups is not statistically significant (Figure 3B). 

Based on these data, one can conclude that the administration of S-15176 partially reversed ultrastructural changes in liver mitochondria during the development of diabetes mellitus.

### 3.3. Effect of S-15176 Difumarate Salt on Diabetes-Induced Changes in the Expression Level of Proteins Responsible for Mitochondrial Biogenesis and Mitochondrial Dynamics

The quantification of mitochondrial DNA (mtDNA) copies is considered to be an indirect way to evaluate the number of mitochondria per cell. As seen in Figure 4, the development of T2DM results in a decrease in the number of mtDNA copies in liver tissue by 15%.

The administration of S-15176 to control animals has no a statistically significant effect on the mtDNA level. Similar to the number of liver mitochondria per plate (Figure 3B), the mtDNA content in the liver tissue of S-15176-treated T2DM mice tends to increase (Figure 4). In parallel, the T2DM + S-15176 group shows a two-fold increase in the expression level of *Ppargc1a* encoding the PGC-1α protein, which mediates mitochondrial biogenesis (Figure 5A). It is interesting that, in the course of the development of T2DM, the level of *Ppargc1a* expression tends to increase.

Figure 5B–D show the data on the relative expression level of genes encoding the proteins responsible for the processes of mitochondrial fission (*Drp1*) and fusion (*Mfn2* and *Opa1*). One can see that the content of *Mfn2* and *Opa1* mRNA in diabetic mouse liver is slightly decreased, and that of *Drp1* mRNA is slightly increased. In the liver of T2DM mice treated with S-15176, the expression profile of *Drp1* recovers to that in the control. The content of *Mfn2* and *Opa1* mRNA in the T2DM + S-15176 group does not differ significantly from the T2DM group. It is should be noted that in the CTR + S15176 group, the level of *Opa1* mRNA is decreased by 22%, and the content of *Mfn2* mRNA tends to decrease compared to that in the CTR group.

Figure 5E,F presents the data on the relative expression level of *Pink1* and *Parkin*, which are responsible for autophagic clearance of mitochondria. As can be seen, the content of *Parkin* mRNA in the liver of T2DM mice is decreased by 30%. Treatment of diabetic mice with S-15176 does not significantly affect the expression profiles of *Pink1* and *Parkin*. It should be noted that in the CTR + S15176 group, the expression level of *Parkin* is reduced by 23% compared to the control.

In T2DM, the relative level of expression of *Cpt1a* that encodes carnitine palmitoyltransferase 1A, which is responsible for fatty acid β-oxidation in liver mitochondria, is increased 1.45-fold. The treatment of diabetic animals with S-15176 decreases the mRNA content of *Cpt1a* 1.25-fold (Figure 5G). 

### 3.4. Effects of T2DM and S-15176 on the Functioning of Mouse Liver Mitochondria

Table 3 shows the respiration rates of mouse liver mitochondria at different functional states. One can see that the T2DM group exhibits an increased rate of the resting (state 4) respiration of isolated mitochondria. The respiratory control ratio (RCR), which indicates the effectiveness of mitochondria in promoting oxidative phosphorylation, in the T2DM group is lower by ~20% than that in the control. Treatment of diabetic mice with S-15176 leads to an increase in the rates of mitochondrial respiration in states 3 and 4. The RCR index in the T2DM + S-15176 group is higher by 13% than that in the T2DM group, but this difference is not statistically significant. It should be noted that in S-15176-treated control mice, the RCR index slightly (statistically insignificant) decreases.

The mouse liver mitochondria of the experimental groups were assayed for oxidative injury. Lipid peroxidation was quantified by measuring the accumulation of thiobarbituric reactive substances (TBARS). As shown in Figure 6, the TBARS concentration in liver mitochondria of T2DM mice is 1.4 times higher than that in control animals. The concentration of TBARS in the T2DM + S-15176 group decreases as compared with that in diabetic mice and reaches the level in control animals.

## 4. Discussion

In this work, we have examined the effects of S-15176, a potent derivative of the anti-ischemic agent trimetazidine, on the ultrastructure and functions of liver mitochondria in control and diabetic animals (the mouse model of high fat/streptozotocin-induced T2DM). It is well established that mitochondria of the liver are one of the main intracellular targets in the course of the development of T2DM [3,4]. The trimetazidine derivative S-15176 was reported to have diverse effects on mitochondria: the compound inhibits CPT-1, exhibits uncoupling and antioxidant properties, and suppresses the MPT pore opening [19,20,21,22,23]. Based on the above, we hypothesized that this compound would affect the mitochondrial function during the development of type 2 diabetes.

It is should be noted that, despite the known positive effects of S-15176 as an anti-ischemic and antioxidant agent, we observed that the administration of S-15176 led to a suppression of mitochondrial fusion and an increase in mitochondrial fission in the liver of control mice (Figure 5). There was also a slight decrease in the expression level of Parkin that mediates the clearance of damaged mitochondria via mitophagy pathway (Figure 5F). Thus, one can conclude that the drug administration leads to changes in mitochondrial dynamics and mitophagy under normal conditions.

To investigate the effects of S-15176 on the development of mitochondrial dysfunction associated with diabetes, we used a standard mouse model for T2DM (high-fat diet + low doses of streptozotocin). The successful induction of T2DM in mice was confirmed by the IPGTT and the IPIST tests; diabetic mice showed hyperglycemia and insulin resistance. 

Electron microscopy studies revealed that T2DM induced the swelling and hypertrophy of liver mitochondria, as well as the vacuolation and disintegration of mitochondrial cristae accompanied by the loss of components of the matrix of the organelles. In parallel, the processes of mitochondrial fission enhanced, which manifested themselves as the appearance of micro-mitochondria (about 5% of the mitochondrial population) in liver cells. Moreover, the expression level of *Drp1* slightly increased and that of *Opa1* and *Mfn2* somewhat decreased. Altogether, these facts point to the fragmentation of the mitochondrial network in liver tissue upon the development of diabetes. In parallel, the expression level of *Parkin* decreased, which indicates the suppression of mitophagy. The observed changes in the expression of genes involved in the regulation of mitochondrial dynamics and mitophagy are consistent with the literature data [34], which suggests that the used mouse model successfully reproduces the pathology of diabetes mellitus. 

It was also found that T2DM led to a significant decline in the number of mtDNA copies in the mouse liver tissue. It is interesting that, in the liver, the expression level of the gene encoding the master regulator of mitochondrial biogenesis, peroxisome proliferator-activated receptor-c coactivator 1α (PGC-1α), was slightly increased. At the same time, mitochondrial biogenesis is considered to be suppressed in many tissues upon the development of diabetes. However, a number of studies reported on the activation of this process in the liver (see review [34]), which may be associated with the important role of this organ in detoxification.

T2DM-induced pathological changes in the mitochondrial ultrastructure were accompanied by disorders of mitochondrial function. Firstly, oxidative damage to liver mitochondria increased. As can be seen in Figure 6, the induction of T2DM resulted in the elevated level of lipid peroxides, which correlates with the literature data [3,4]. Secondly, a decline in the efficiency of ATP synthesis in diabetic animals was observed. As can be seen from Table 3, the respiratory control ratio of liver mitochondria of diabetic mice was significantly reduced due to an increase in the state 4 (non-ADP-stimulated) respiration rate. On the one hand, this can be explained by the destabilization of the mitochondrial respiratory chain supercomplexes, which occurs in T2DM [35]. On the other hand, the development of T2DM was found to be accompanied by an increase in the level of free fatty acids and the expression of uncoupling proteins (UCPs). Importantly, UCP expression can be an adaptive response to elevated ROS generation by mitochondria [35,36].

After we found the above changes in mouse liver mitochondria in experimental T2DM, we analyzed the effects of S-15176 at a low dose (1.5 mg/kg/day i.p. for 20 days) on the ultrastructure and functions of the organelles.

Since S-15176 at the dose applied inhibits the activity of CPT-1 [20], one can suggest that the drug is able to exhibit hypoglycemic properties. Here, we showed that the expression level of the gene encoding this mitochondrial protein is slightly increased during the development of T2DM, which corresponds to the literature data [37,38]. After the treatment of T2DM mice with S-15176, the tendency of *Cpt1a* expression to decline was clearly seen. However, no hypoglycemic effect of S-15176 at the concentration used was found. Meanwhile, this dose of S-15176 partially reversed the disorders in mitochondria from the liver tissue of T2DM mice. 

The administration of S-15176 to diabetic mice prevented some of the T2DM-induced ultrastructural alterations in liver mitochondria. The average size of the organelles decreased almost to the levels of control animals. Mitochondria of S-15176-treated T2DM mice, in contrast to the organelles of diabetic animals, was demonstrated to be in close contact with endoplasmic reticulum. It should be noted that these changes can be responsible for the enhanced mitochondrial biogenesis. Indeed, S-15176-treated T2DM mice showed an overexpression of the gene encoding PGC-1α protein (almost two-fold) and a restoration of the expression level of *Drp1* in the liver to the level in control animals (Figure 5). Altogether, our results indicate that the administration of S-15176 stimulates the processes of mitochondrial biogenesis. Earlier, the activation of mitochondrial biogenesis was revealed during the trimetazidine-treated skeletal muscle dysfunction and wasting occurring in cancer cachexia [39]. At the same time, the administration of S-15176 did not affect the decrease in expression levels of *Parkin* in the liver of diabetic animals, but led to a slight decrease in those of control mice. This suggests that the drug inhibits mitophagy pathway, which may not always lead to negative changes in patologies. Recently, it was found that trimetazidine protects against myocardial ischemic injury by inhibition of excessive autophagy [40].

Along with enhancement of biogenesis of mitochondria and the preservation of their structure, S-15176 partially restored the bio-energetic function of mouse liver mitochondria in T2DM. The administration of S-15176 to diabetic animals resulted in an increase in the rates of mitochondrial respiration in the states 3 and 4. This may be due to both an increase in the content of respiratory chain complexes as a result of the S-15176-induced activation of mitochondrial biogenesis and a known mild uncoupling effect of the compound. The respiration control ratio in the T2DM + S-15176 group tended to restore. One can suggest that the effect of S-15176 on mitochondrial dysfunction in the liver may be more pronounced when taking into account the zone-dependent heterogeneity of hepatocytes. As is known, in the liver of rodents and humans, three metabolic zones of equal width, extending from the portal area to the central vein, are distinguished: periportal (zone 1), intermediate (zone 2), pericentral (zone 3). Zone 3 was found to be most susceptible to morphological and functional changes in a number of pathologies [41]. Since we isolated mitochondria from the entire mouse liver, it is not possible to assess the effect of S-15176 on the parameters of mitochondrial respiration in discrete functional areas of the organ.

The stimulation of mitochondrial respiration is known to be one of the factors responsible for the suppression of mitochondria-mediated overproduction of ROS [5,35]. Indeed, the treatment of diabetic mice with S-15176 significantly lowered the concentration of malondialdehyde in liver mitochondria, suggesting that the drug prevented the induction or development of oxidative stress in T2DM. This effect of S15176 may also be related to the direct action of this compound on mitochondria. In in vitro experiments on isolated liver mitochondria, S-15176 was shown to induce a mild uncoupling [23], which may lead to the suppression of excessive generation of ROS by mitochondria.

It should be noted that S-15176 at the dose used had no a statistically significant effect on some alterations in the functioning of liver mitochondria in diabetic animals. Notably, S-15176-treated T2DM mice presented only a tendency to restore the number of mitochondria, the content of mtDNA, and the expression level of *Mfn2* in liver tissue. S-15176-treated and non-treated T2DM animals contained a population of liver micro-mitochondria less than 1 μm in size, which were not detected in the control group. This may indicate that S-15176 treatment did not affect the T2DM-related fragmentation of the mitochondrial network. 

## 5. Conclusions

The results obtained demonstrate that S-15176 at the dose used has a complex effect on liver mitochondria. The mechanisms of modulatory effect of S-15176 on pathological changes in the structure and functioning of liver mitochondria in T2DM can be related to increase in mitochondrial biogenesis and the inhibition of lipid peroxidation in the organelles. At the same time, S-15176 at the dose used had no a statistically significant effect on a number of main functional parameters of liver mitochondria of diabetic animals. In this regard, we cannot exclude the possibility that in the case of a longer course of treatment and a higher dose of the drug, the protective effect of S-15176 against hepatic mitochondrial dysfunction in diabetes would be more pronounced.

## Figures and Tables

**Figure 1 biology-09-00309-f001:**
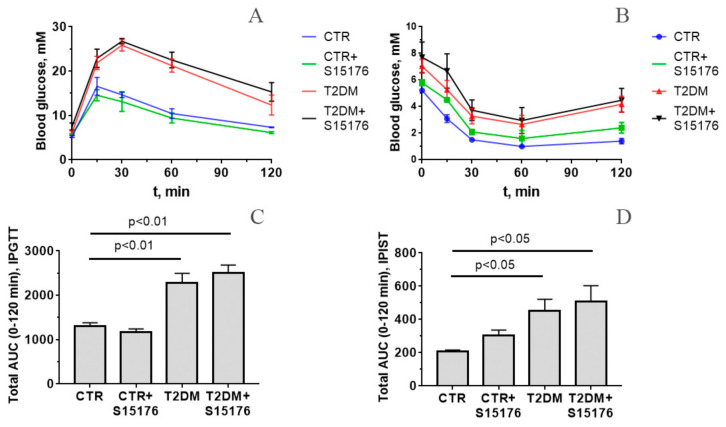
Altered levels of blood glucose as indicated by (**A**) intraperitoneal glucose tolerance test (IPGTT) in control (CTR), control + S-15176 (CTR+S-15176), diabetic (T2DM), and S-15176-treated diabetic (T2DM + S-15176) mice; (**B**) intraperitoneal insulin sensitivity test (IPIST) in the experimental group of animals; (**C**,**D**) the total area under the curve (AUC) during IPGTT and IPIST in the experimental groups of animals. The differences were considered statistically significant at *p* < 0.05. The values are given as mean ± SEM of the number of independent experiments indicated (*n* = 3–4).

**Figure 2 biology-09-00309-f002:**
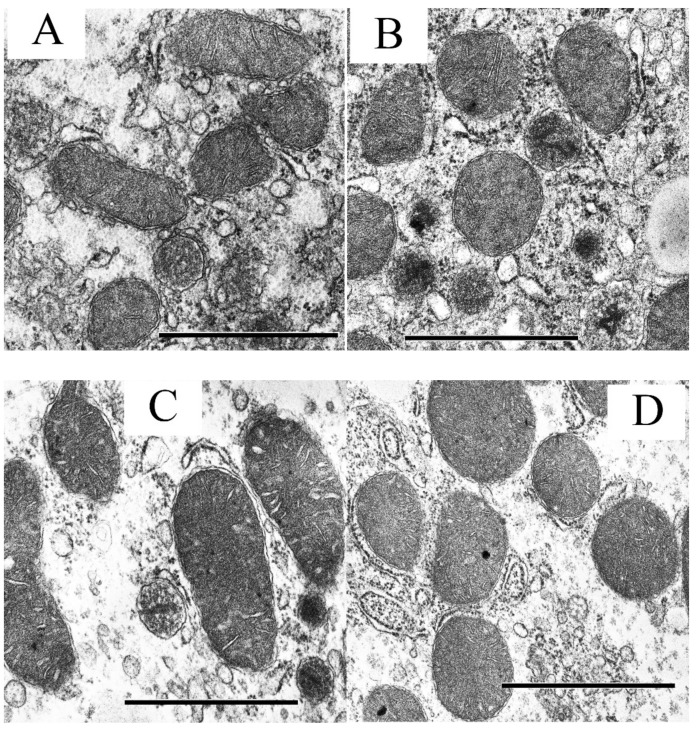
Transmission electron micrographs of typical liver mitochondria of mice of four experimental groups: CTR (**A**), CTR + S-15176 (**B**), T2DM (**C**,**E**) and T2DM + S-15176 (**D**,**F**). Samples from two livers were analyzed in each experimental group. The number of examined fields of view (25 μm^2^) in the groups varies from 15 to 25. The number of mitochondria analyzed in each group varies from 100 to 200. The bar is equal to 1 μm (**A**–**D**) or 0.5 μm (**E**,**F**).

**Figure 3 biology-09-00309-f003:**
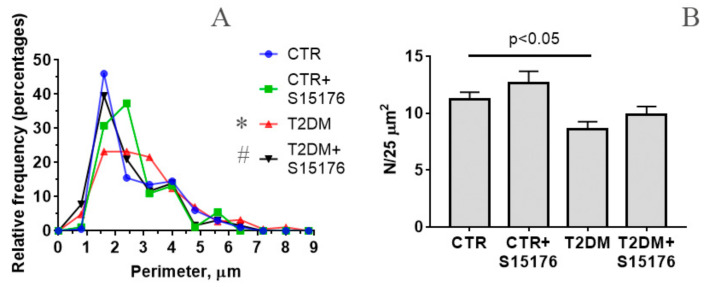
Morphometric parameters of liver mitochondria from experimental mice: (**A**) a histogram of distribution of the mitochondrial perimeter in the groups. The number of mitochondria analyzed in each group varies from 100 to 200. Statistical significance was estimated by the Kruskal-Wallis test: * *p* < 0.05 compared to the CTR group. # *p* < 0.05 compared to the T2DM group; (**B**) the number of mitochondria per field of view (25 μm^2^). The number of examined fields of view (25 μm^2^) in the groups varies from 15 to 25. The differences were considered statistically significant at *p* < 0.05.

**Figure 4 biology-09-00309-f004:**
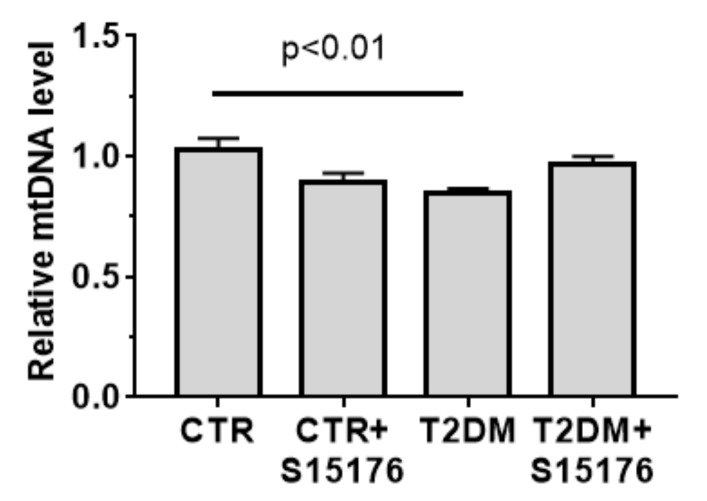
The relative mtDNA levels in the liver of experimental animals. A real-time qPCR was carried out to determine the mtDNA copy number, which is calculated as the ratio of mitochondrial DNA (*ND4*) to nuclear DNA (*GADPH*) (*n* = 5–6). The differences were considered statistically significant at *p* < 0.05.

**Figure 5 biology-09-00309-f005:**
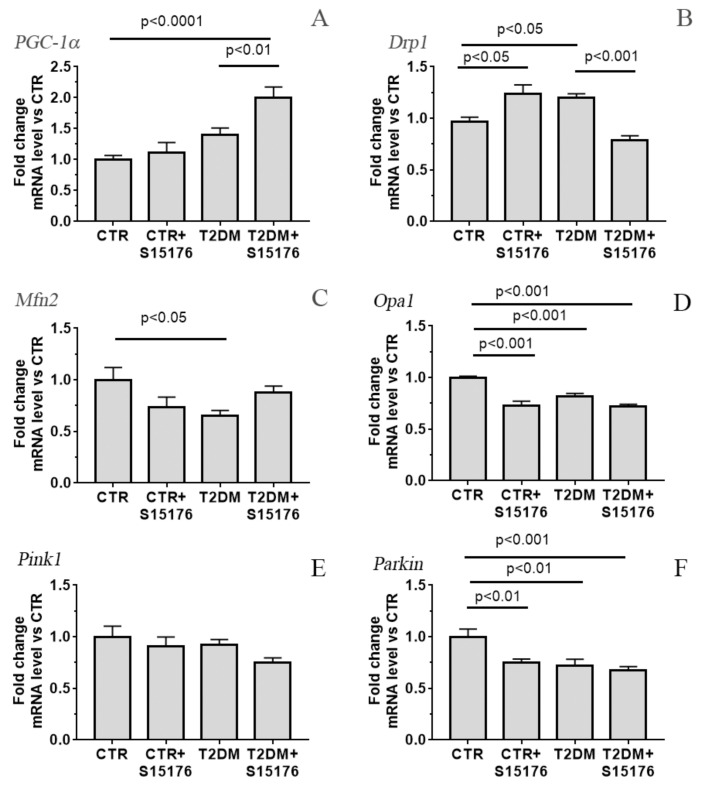
The mRNA levels of *Ppargc1a* (**A**) *Drp1* (**B**), *Mfn2* (**C**), *Opa1* (**D**), *Pink1* (**E**), *Parkin* (**F**), and *Cpt1a* (**G**) in the liver of experimental animals (*n* = 5–6). The differences were considered statistically significant at *p* < 0.05.

**Figure 6 biology-09-00309-f006:**
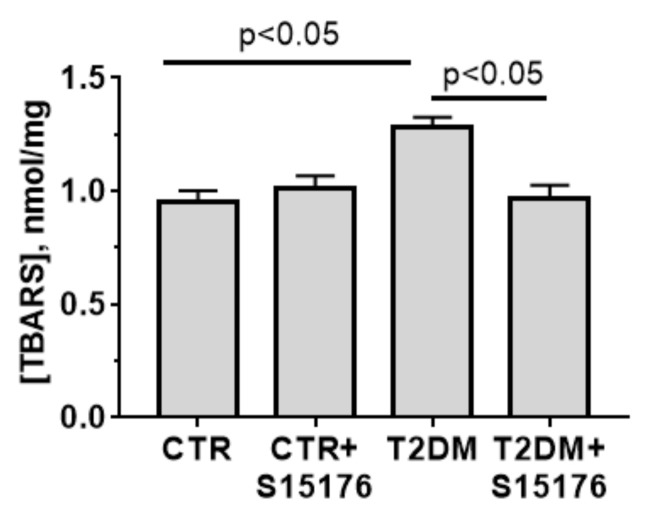
S-15176 significantly suppresses the T2DM-induced lipid peroxidation in mouse liver mitochondria. Lipid peroxidation was assessed by the level of TBARS (MDA and other minor aldehyde species) in mitochondria isolated from the liver of mice from the experimental groups. Values are given as means ± SEM (*n* = 5). The differences were considered statistically significant at *p* < 0.05.

**Table 1 biology-09-00309-t001:** List of gene-specific primers for the real-time PCR analysis.

Gene	Forward (5′→3′)	Reverse (5′→3′)
*Nd4*	ATTATTATTACCCGATGAGGGAACC	ATTAAGATGAGGGCAATTAGCAGT
*Gadph*	GTGAGGGAGATGCYCAGTGT	CTGGCATTGCTCTCAATGAC
*Drp* *1*	TTACAGCACACAGGAATTGT	TTGTCACGGGCAACCTTTTA
*Mfn2*	CACGCTGATGCAGACGGAGAA	ATCCCAGCGGTTGTTCAGG
*Ppargc1a*	CTGCCATTGTTAAGACCGAG	GTGTGAGGAGGGTCATCGTT
*Cpt1a*	GGCATAAACGCAGAGCATTCCTG	CAGTGTCCATCCTCTGAGTAGC
*Pink1*	TTGCCCCACACCCTAACATC	GCAGGGTACAGGGGTAGTTCT
*Prkn*	AGCCAGAGGTCCAGCAGTTA	GAGGGTTGCTTGTTTGCAGG
*Opa1*	GGACCCAAGAGCAGTGTGTT	CGAGACTCCAGGTTCTTCCG
*Rplp2*	CGGCTCAACAAGGTCATCAGTGA	AGCAGAAACAGCCACAGCCCCAC

**Table 2 biology-09-00309-t002:** Animal weights and biochemical characteristics in the studied groups.

	CTR	CTR + S15176	T2DM	T2DM + S15176
Initial BW, g	16.4 ± 1.0	15.6 ± 1.1	14.5 ± 0.8	16.1 ± 0.9
Final BW, g (after two months)	25.2 ± 0.8 (↑ 153.7%)	25.3 ± 1.0 (↑ 161.9%)	24.5 ± 0.6 * (↑ 172.4%) *	25.5 ± 0.6 (↑ 158.4%)
BG (fed state)	9.7 ± 0.6	9.3 ± 0.6	20.0 ± 2.2 *	20.6 ± 1.4 *

Values are given as mean ± SEM of five independent experiments. BW, body weight; BG, blood glucose. The weight gain relative to the initial values is indicated in parentheses. * *p* < 0.05 compared to the control group (CTR).

**Table 3 biology-09-00309-t003:** The respiration rates of mouse liver mitochondria of the experimental groups.

Group	V Respiration, nmol O_2_ × min^−1^ × mg^−1^ Protein			RCR
	State 2	State 3	State 4	
CTR	22.0 ± 1.7	82.6 ± 1.9	19.9 ± 0.9	4.2 ± 0.2
CTR+ S-15176	19.5 ± 2.2	85.2 ± 4.1	24.2 ± 2.1	3.5 ± 0.3
T2DM	22.5 ± 1.3	85.4 ± 8.3	26.1 ± 1.5 *	3.4 ± 0.2 *
T2DM+S-15176	19.4 ± 0.9	110.1 ± 4.4 *#	28.4 ± 1.1 *	3.9 ± 0.2

Medium composition: 130 mM KCl, 5 mM NaH2PO4, 10 µM EGTA, and 10 mM HEPES-KOH, pH 7.4. Respiration of isolated mitochondria was fueled by 2.5 mM glutamate and 2.5 mM malate. Mitochondrial respiration in state 3 was initiated by 200 µM ADP. The results are presented as means ± SEM (n = 4). * *p* < 0.05 compared to the control group (CTR). # *p* < 0.05 compared to the T2DM group.

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
