# Peer review of "The Effect of S-15176 Difumarate Salt on Ultrastructure and Functions of Liver Mitochondria of C57BL/6 Mice with Streptozotocin/High-Fat Diet-Induced Type 2 Diabetes"

_biology, 2020, doi:10.3390/biology9100309_

Round 1

Reviewer 1 Report

The authors made efforts to answer even if they did not perform some expetiments.Data obtained are not indicative of a strong protective role of S-15176 on  mitochondria in diabetes but suggest a regulation of mitochondriogenesis. Probably, higher doses might result more effective. However the revised manuscript is adequately written and technically well performed. In my opinion the study deserves to be published.

Reviewer 2 Report

My questions had been well addressed. This submission is acceptable

Reviewer 3 Report

I accept the revised manuscript for publication.

Reviewer 4 Report

In the previous version of the paper some data concerning mitochondrial fission/fusion were confusing and it was not clear what these were based on.  The authors have now removed this data and also done some additional experiments that further support their findings. Thus, I am happy with the improvement of the paper and now find the paper appropriate for publication.

This manuscript is a resubmission of an earlier submission. The following is a list of the peer review reports and author responses from that submission.

Round 1

Reviewer 1 Report

The authors have tested the ability of S-15176, an anti-oxidant, anti-ischemic agent that inhibits mitochondrial permeability transition, to ameliorate mitochondrial changes in diabetic mice. They show that high fat diet induced diabetes in mice led to glucose intolerance and ultrastructural changes of liver mitochondria. The S-15176 treatment did not affect blood glucose levels in the mice, nor the responses to glucose or insulin. However, the treatment normalized the morphology of the mitochondria,  according to the  morphometric parameters (figure 3).  However, based on this data the treatment also induces morphological changes in the mitochondria of the control mice, although not in the same direction than in the diabetic mice.

Based on PGC1a expression, the treatment also seems to induce mitochondrial biogenesis, which is already up in the diabetic mice.

The authors clearly show that the treatment induces some changes in the mitochondria, and normalizes some organellar defects in the diabetic mice, however these do not seem to bear functional significance as no changes in the blood glucose or in the insulin sensitivity are seen. In the conclusions (row 465-466) the authors state that the treatment had no effect on the healthy liver. However, this does not seem to be completely true, as clearly several changes are seen, although they often tend to be in different direction than the effect in the diabetic mice.

I have few questions about the data. Firstly, the data on the fission and fusion is a bit confusing. Just looking at the expression of genes involved in fission and fusion, the treatment seems to affect these differently in the control and diabetic mice. Based on the Drp1 and Mfn2 expression, diabetes induces fission, while the treatment reduces it in diabetic mice and induces it in control mice. This seems to be also in line with the  morphological results in figure 3. In the rows 330-331 the authors then state that fission is increased in diabetic mice, but it is not very clear what this statement is based on. They then show some numerical data (rows 342-346) that suggest that fission is increased both upon diabetes and upon treatment. However, the authors show nowhere, where this data is coming from. I assume this is from the number drp labeled mitochondria from the EM images but how this is quantified, how many images, how many mitochondria are quantified.

On row 199 The authors state that the weight gain in T2DM mice is significantly higher, however, these mice were the smallest in the beginning and still remain the smallest at the end of the experiment, so eventhough they show relatively higher weight gain, they are not really obese or bigger that the other groups

In Table 2 what does BG mean? This is missing from the abbreviations and also the unit is missing.If it means blood glucose, the results seem to be somewhat different from those in figure 1. Are these in fed state (10-20) and figure 1 starting values (5-8) the fasting values? This should be clarified more.

I would like to see speculation on the mechanism of the drug, the authors suggest anti-oxidative effects, suppression of MPT opening or suppression of beta-oxidation. What do they think is behind the mitochondrial morphology improvement in these mice?  They further suggest that it could be mitochondrial biogenesis that improved mitochondrial function. It would be fairly easy to measure for example the ROS levels and the mitochondrial mass to see if either one of these is behind the improvement.

Minor comments on the methods:

In the methods section the primers from the control Rplp2 are missing from the primer table.

For the mtDNA copy number assessment the authors have used total DNA extracted by a kit (DNA-Extran 2, sitol). In general it is not recommended to use any column based method for DNA extraction for these assays. The column separation is always based on size and either favors large nuclear DNA or small plasmid like mtDNA, while depleting majority of the other one. I am not familiar with this kit and was not able to find the principle it is based on anywhere, but most kits use column separation.

Author Response

Dear Reviewer,

Thank you very much for the detailed review of our work and valuable recommendations. We have checked the manuscript and made the necessary corrections. The answers to the questions your raised are provided below.

Comments: Firstly, the data on the fission and fusion is a bit confusing. Just looking at the expression of genes involved in fission and fusion, the treatment seems to affect these differently in the control and diabetic mice. Based on the Drp1 and Mfn2 expression, diabetes induces fission, while the treatment reduces it in diabetic mice and induces it in control mice. This seems to be also in line with the  morphological results in figure 3. In the rows 330-331 the authors then state that fission is increased in diabetic mice, but it is not very clear what this statement is based on. They then show some numerical data (rows 342-346) that suggest that fission is increased both upon diabetes and upon treatment. However, the authors show nowhere, where this data is coming from. I assume this is from the number drp labeled mitochondria from the EM images but how this is quantified, how many images, how many mitochondria are quantified.

Answer: We agree with your comment and have made the appropriate changes to the revised version of the manuscript. Indeed, our studies showed an increase in the expression level of Drp1 and a decrease in the expression of Mfn2 in the liver of diabetic animals as compared to control animals. These data are consistent with the results of morphometric analysis of mitochondria in liver tissue obtained using electron microscopy. The administration of S15176 to diabetic animals normalized both the expression of DRP1 and the size of mitochondria in hepatocytes. The micrograph of Drp-1 labeling mitochondrion (Fig. 6) and the data on the number of DRP1-labeled mitochondria were obtained on a suspension of isolated mitochondria and therefore may be incorrect. Taking into account your remark, we have removed Figure 6 and its description from the manuscript.

Comments: On row 199 The authors state that the weight gain in T2DM mice is significantly higher, however, these mice were the smallest in the beginning and still remain the smallest at the end of the experiment, so eventhough they show relatively higher weight gain, they are not really obese or bigger that the other groups

Answer: Indeed, our results and literature data indicate that mice with streptozotocin/high-fat diet-induced T2DM are characterized by an increase in weight gain, which is not very significant. According to the literature (O'Brien PD, Guo K, Eid SA, et al. Integrated lipidomic and transcriptomic analyses identify altered nerve triglycerides in mouse models of prediabetes and type 2 diabetes. Dis Model Mech. 2020;13(2):dmm042101. DOI:10.1242/dmm.042101), this experimental model of T2DM does not imply a significant increase in animal weight. When fed a high-fat diet, the mice gained weight, but the subsequent administration of streptozotocin led to a partial decrease in the animal's body weight. In this regard, we have mentioned that upon the induction of T2DM, mice gained more weight than in the control.

Comments: In Table 2 what does BG mean? This is missing from the abbreviations and also the unit is missing.If it means blood glucose, the results seem to be somewhat different from those in figure 1. Are these in fed state (10-20) and figure 1 starting values (5-8) the fasting values? This should be clarified more.

Answer: Sorry for this; the abbreviation (BG - blood glucose (fed state)) has been deciphered. Table 2 presents the data on blood glucose levels of fed animals. To perform the glucose tolerance test (Figure 1), we used starving animals. Corresponding clarifications have been added to the revised version of the manuscript.

Comments: I would like to see speculation on the mechanism of the drug, the authors suggest anti-oxidative effects, suppression of MPT opening or suppression of beta-oxidation. What do they think is behind the mitochondrial morphology improvement in these mice?  They further suggest that it could be mitochondrial biogenesis that improved mitochondrial function. It would be fairly easy to measure for example the ROS levels and the mitochondrial mass to see if either one of these is behind the improvement.

Answer: Thank you for this comment. As shown in our work, treatment of diabetic animals with S15176 lead to an increase in mitochondrial biogenesis at the cellular level. We suggest that this mechanism may underlie the improvement in mitochondrial morphology. In parallel, S15176 decreases the level of the key marker of lipid peroxidation malondialdehyde in mitochondria, which may prevent oxidative damage to these organelles in diabetes. In addition, some studies demonstrated the direct effect of this compound on mitochondria. In experiments on isolated liver mitochondria, S15176 was shown to induce a mild mitochondrial uncoupling, which may lead to the suppression of excessive generation of ROS by mitochondria in pathologies. Corresponding changes were made to the Discussion section.

Minor comments: on the methods: In the methods section the primers from the control Rplp2 are missing from the primer table.

Answer: The primers from the control Rplp2 have been added in Table 1.

Minor comments: For the mtDNA copy number assessment the authors have used total DNA extracted by a kit (). In general it is not recommended to use any column based method for DNA extraction for these assays. The column separation is always based on size and either favors large nuclear DNA or small plasmid like mtDNA, while depleting majority of the other one. I am not familiar with this kit and was not able to find the principle it is based on anywhere, but most kits use column separation.

Answer: Thank you for this comment. The catalog number EX-511 of the DNA-Extran 2 kit (Sintol, Russia) has been added to the appropriate paragraph in the Materials and Methods section. According to the manufacturer's instruction, the set of reagents “DNA-Extran-2” for DNA extraction from animal and human tissues (Cat. No. EX-511, ZAO Sintol Company, Moscow, Russia) is used for on sequential processing of the sample with solutions for cell lysis (overnight at 56 C) and precipitation of proteins and DNA using precipitation solutions and centrifugation. The advantages of this kit are the complete extraction of DNA from cells, which minimizes DNA loss and fragmentation during purification. The isolated DNA has a high level of purity (A260/280 = 1.8-1.9) and is suitable for the PCR analysis, hybridization and other studies. It should be noted that this method has been used in a number of studies (see the list below).

Spangenberg, V., Arakelyan, M., Cioffi, M.d.B. et al. Cytogenetic mechanisms of unisexuality in rock lizards. Sci Rep 10, 8697 (2020). https://doi.org/10.1038/s41598-020-65686-7

S.V. Mezhzherin, L. M. Yanovich, , E. I. Zhalay, , L. A. Vasilieva, M. M. Pampura. Genetic and morphological variability and differentiation of freshwater mussels (bivavia, unionidae, anodontinae) in Ukraine. Vestnik zoologii, 48(2): 99–110, 2014. DOI: 10.2478/vzoo-2014-0011

Reviewer 2 Report

The authors adopted a dietary model plus low-streptozotocin to induce type 2 diabetes in mice and analysed the effects of S-15176 on liver mitochondria. Multiple morphological, biochemical and molecular methods were performed to characterize mitochondrial events. The main conclusion is that the drug was able to recover mitochondrial size and function, but in my opinion much more experiments are needed to clearly demonstrate this point.

Major changes are suggested to ameliorate this study as follows:

1.Mitochondrial dynamic and adaptation to more energy intake in the diabetic mice liver must be better discussed and analysed. Opa1 expression a recognized marker of cristae maintenance is lacking.

2.Mitophagy, generally useful to clean aberrant and dysfunctional mitochondria, must be checked by PINK1-PARKIN or TEM in zone 3. See Ma et al Cells 9, 837, 2020. The authors hypothesized this event only.

3.The authors measured functional parameters in extracted mitochondria, but in the liver, there are three different metabolic zones with striking metabolic differences that must be characterized on sections (Stacchiotti et al. PlosOne 11, e0148115, 2016).

  1. ER stress response must be assessed together with ER-mitochondria juxtapositions at TEM.
  2. Unfortunately, crucial parameters like mtDNA content, Mf2 expression, number of mitochondria, RCR index in diabetic or diabetic treated with S-15176 compound were not statistically significant. Probably, a higher dose or more time than 20 days of treatment, might be more indicative.

Minor changes:

1. Line 44- Please insert References.

  1. Line 78- Drp1 and PGC1-alpha must be introduced with a dedicated sentence.

3. Line 108-Indicate a Reference or the reason to choose the S-15176 dose in this study.

Author Response

Dear Reviewer,

Thank you very much for the detailed review of our work and valuable recommendations. We have checked the manuscript and made the necessary corrections. The answers to the questions your raised are provided below.

Comments: 1.Mitochondrial dynamic and adaptation to more energy intake in the diabetic mice liver must be better discussed and analysed. Opa1 expression a recognized marker of cristae maintenance is lacking.

Answer: Thank you for this comment. We have performed additional experiments and determined the expression level of Opa1 in the experimental groups. The necessary changes were made to the Results and Discussion sections of the revised version of our manuscript.

Comments: 2.Mitophagy, generally useful to clean aberrant and dysfunctional mitochondria, must be checked by PINK1-PARKIN or TEM in zone 3. See Ma et al Cells 9, 837, 2020. The authors hypothesized this event only.

Answer: We have performed additional experiments and examined the expression level of PINK1-PARKIN. The necessary changes were made to the Results and Discussion sections of the revised version of our manuscript.

Comments: 3.The authors measured functional parameters in extracted mitochondria, but in the liver, there are three different metabolic zones with striking metabolic differences that must be characterized on sections (Stacchiotti et al. PlosOne 11, e0148115, 2016).

Answer: Thank you for your comment. We have made the appropriate corrections and added the literary reference in the Discussion section of the revised version of our manuscript. Indeed, one can suggest that the effect of S-15176 on the development of diabetes-induced mitochondrial dysfunction may be more pronounced in the pericentral zone 3 of the liver. Since we isolated mitochondria from the entire mouse liver, it is not possible to assess the effect of S-15176 on mitochondrial respiration depending on the hepatic zone. It should be noted significant amounts of tissue are required for the isolation of mitochondria and the subsequent assessment of mitochondrial respiration. Traditional methods of isolation of mitochondria described in the available literature do not yet allow obtaining mitochondrial suspensions from three zones of the liver of mice. We will take into account your remark and try to investigate this issue in our future work.

Comments: 4. ER stress response must be assessed together with ER-mitochondria juxtapositions at TEM.

Answer: The present work was aimed at a comprehensive study of the morphology and functions of mitochondria in diabetes and the effect of S-15176 treatment on pathological changes in these organelles. Our observation that T2DM reduces ER-mitochondria interactions was confirmed earlier by many studies using TEM analysis (Rieusset J. Cell Death Dis. 2018;9(3):388, doi:10.1038/s41419-018-0416-1; Tubbs E, et al. Diabetes. 2014; 63:3279–3294, doi: 10.2337/db13-1751; Theurey P, et al., 2016;8:129–143, doi: 10.1093/jmcb/mjw004; Leem J, Koh EH. Exp Diabetes Res. 2012;2012:242984, doi:10.1155/2012/242984; Thivolet С. et al., Plos One, 2017, doi:10.1371/journal.pone.0182027). We think that ER stress response in diabetes is a hot and separate topic in research and should be analyzed in detail using several tools. We will try to pay more detailed attention to this issue in our future works.

Comments: 4.Unfortunately, crucial parameters like mtDNA content, Mf2 expression, number of mitochondria, RCR index in diabetic or diabetic treated with S-15176 compound were not statistically significant. Probably, a higher dose or more time than 20 days of treatment, might be more indicative.

Answer: We agree with this comment. Indeed, S15176 at the dose used had no a statistically significant effect on a number of functional parameters of liver mitochondria of diabetic animals. Meanwhile, we consider it possible to publish this work so that other researchers use this data for further studies to develop new regimens for the drug administration. It should be noted that at the beginning of the study, we were faced with the question of whether this compound, like other CPT1 inhibitors and mitochondria-targeted agents, is capable of suppressing the development of diabetes mellitus. To test this, we chose the dose of S-15176 that inhibits CPT1. Based on the results obtained, we cannot exclude the possibility that in the case of a longer course of treatment and a higher dose of the drug, the protective effect of S-15176 against hepatic mitochondrial dysfunction in diabetes would be more pronounced.

Minor changes: Line 44- Please insert References.

Answer: The reference has been added.

Minor changes: Line 78- Drp1 and PGC1-alpha must be introduced with a dedicated sentence.

Answer: All concerns have been corrected.

Minor changes: Line 108-Indicate a Reference or the reason to choose the S-15176 dose in this study.

 Answer: The dose selection was based on the available data on effect of the drug on the activity of carnitine palmitoyltransferase I. At this dose, the drug inhibited the activity of the mitochondrial enzyme in the liver. The corresponding reference has been added.

Reviewer 3 Report

This study revealed that intervention with S-15176 had no hypoglycemic effect and did not affect the insulin resistance of mice with experimental T2DM. The beneficiary effect of S-15176 was linked to the partial preservation of mitochondrial structure and functions in the mouse liver during T2DM. A revision is suggested.

  1. The data presentation needs to be improved, for example, in fig2, different figure sizes are found. Fig2D is the smallest. Scale bars are different in each photo.
  2. The animal numbers are different in the figures. Please explain.
  3. Fig4 is unclear. p<0.05 means? There are three bars under that Line (p<0.05). S-15176 also reduce mtDNA in NON-DM condition?
  4. Fig5 needs to be further confirmed by the Western blotting.
  5. MDA levels in plasma need to be tested too.

Author Response

Dear Reviewer,

Thank you very much for the detailed review of our work and valuable recommendations. We have checked the manuscript and made the necessary corrections. The answers to the questions your raised are provided below.

 Comments: The data presentation needs to be improved, for example, in fig2, different figure sizes are found. Fig2D is the smallest. Scale bars are different in each photo.

Answer: The electron micrographs have been adjusted to the same size and scale. The revised figures 2D and 2F show individual mitochondria to better illustrate the changes in their ultrastructure.

Comments: The animal numbers are different in the figures. Please explain.

 Answer: There were six mice in the diabetic (T2DM and T2DM+S15176) groups, and five mice in the control groups (Cntrl and Cntrl+S15176). To determine the gene expression, we collected tissue samples (150-200 mg) from each animal. For transmission electron microscopy analysis, we fixed three samples of liver tissue (about 5*5 mm) from two mice from each group. The rest of liver tissue was used to study the functional parameters of isolated mitochondria. However, after sampling for electron microscopy, mouse liver tissue was not always sufficient for the procedure for isolating mitochondria (which were obtained from each animal separately). In this regard, the data in Table 3 were obtained using four animals in each group.

Comments: Fig4 is unclear. p<0.05 means? There are three bars under that Line (p<0.05). S-15176 also reduce mtDNA in NON-DM condition?

Answer: Indeed, the differences were considered statistically significant at P < 0.05, as noted in the Materials and methods section. Fig. 4 shows a trend towards a decrease in the level of mtDNA in the control + S15176 group, but this decrease (about 9%) was not statistically significant in comparison with the control (at P <0.05). Statistical analysis showed a statistically significant difference only between the CTR and T2DM groups at P<0.01, which was shown by the line above (in the same manner as it was done in the other figures).

Comments: Fig5 needs to be further confirmed by the Western blotting.

 Answer: Thank you for your comment. It should be noted that small changes observed from the results of PCR analysis are not always confirmed by Western blotting data. More important for us is that changes in the gene expression of the Drp1 and Mfn2 in the liver of diabetic mice (Fig. 5) are consistent with the data of morphometric analysis of electron microscopy images (Figs. 2 and 3). Moreover, the expression level of Ppargc1a, Drp1, Mfn2, Opa1, Parkin, and Cpt1a in the liver of T2DM mice is fully consistent with the available literature data, which was additionally added to the Discussion section. In this regard, we can conclude that the results in Figure 5 are correct.

Comments: MDA levels in plasma need to be tested too.

Answer: Thank you for the comment. MDA level in plasma and tissues is an important indicator of lipid peroxidation and widely used in many studies of the pathogenesis of T2DM. Earlier, it was found that S-15176 has an antioxidant effect in ischemic liver injury and inhibits the generation of thiobarbituric acid-reactive substances in a concentration-dependent manner (Settaf et al., Journal of Pharmacology, 406, 2000, 281-292, doi:10.1016/S0014-2999(00)00599-9.). The present work was aimed at a study of the ultrastructure and functions of mitochondria in diabetes and the effect of S-15176 treatment on pathological changes in these organelles. In this regard, we have tried to characterize precisely mitochondrial dysfunction. We will definitely take into account your remark and determine this indicator in order to characterize the level of oxidative stress not only in mitochondria, but also in the whole organism in our more detailed studies.

Reviewer 4 Report

See attached file

Author Response

Dear Reviewer,

Thank you very much for the detailed review of our work and valuable recommendations. We have checked the manuscript and made the necessary corrections. The answers to the questions your raised are provided below.

Comments: A major concern of this manuscript is that S-15176 did not result in significant differences in T2DM mice. The clearance of blood glucose or the sensitivity to insulin changed between control and T2DM, but neither group was affected by S-15176. Similarly, morphological changes appear more likely due to T2DM than S-15176. Only figure 5A, 5B (mRNA levels for PGC-1 alpha and Drp1) and 7 (accessing lipid peroxidation) show an effect of S-15176 in T2DM livers. In the heart this drug is an anti-ischemic agent, inhibits mitochondrial permeability transition, and prevents the early step in apoptosis by preventing collapse of the electrochemical gradient across the mitochondrial membrane. The idea is that a shift from fatty acid to glucose oxidation may contribute to anti-ischemic effect. It would have been interesting to see, that S15176 has some of these properties also in the experiments described here. For example, could a different concentration of S-15176 lead to an improved glucose tolerance or insulin sensitivity?

 Answer: Thank you very much for your comment and the very interesting question. Indeed, S-15176 difumarate salt at the dose used had no a statistically significant effect on a number of functional parameters of liver mitochondria of diabetic animals. At the beginning of the study, we were faced with the question of whether this compound, like other inhibitors of carnitine palmitoyl transferase I (CPT1) and mitochondria-targeted agents, is capable of suppressing the development of diabetes mellitus. The choice of the dose of S-15176 (1.5 mg/kg) was based on available studies on the effect of the drug on the activity of carnitine palmitoyl transferase I. At a dose of 1.5 mg/kg, the drug inhibits CPT1. The corresponding reference has been added in the text of the revised manuscript. Despite the fact that there is evidence of an anti-ischemic effect of this drug, we recently found that S-15176 at higher concentrations suppresses the rate of oxygen consumption of mitochondria isolated from rat brain and cellular respiration of the rat cerebral cortex neurons in experiments in vitro. We are now preparing an additional manuscript on this research topic. We consider it possible to publish this work so that other researchers use this data for further studies to develop new regimens for the drug administration. We have made some additional literature references in the Discussion section of the revised manuscript.

Comments: The study relies heavily on electron microscopy but the presented micrographs are of poor quality. The images are granular and cristae are hardly recognizable. This makes it problematic to conclude that mitochondria shown in figure 2D are swollen or not. Comparing 2A and 2B makes me think, that the mitochondria in figure 2B could be swollen. The description of the results hints to different morphological aspects of liver mitochondria, however, these results are not shown. For example, micro-mitochondria (lines 292=299); if there is a change, why not showing images to support the results?

Answer: We have improved the quality of the electron micrographs and corrected the description of the results in this part of the work. The micrographs have been adjusted to the same size and scale. According to the morphometric analysis (Fig. 3A), one can conclude that mitochondria in the CTR (fig.2A) and CTR + S15176 (fig. 2B) groups slightly differ in size, but this difference was statistically insignificant. Figures 2E and 2F show individual mitochondria in order to better illustrate the changes in their ultrastructure.

Comments: Figure 3: are these results obtained from experiments with isolated mitochondria or tissue? If isolated mitochondria, how are the authors account for fragmentation in the analysis of the mitochondria during the isolation protocol?

Answer: Figure 3 demonstrates morphological changes in mitochondria in liver tissue. A corresponding clarification has been added to the title of the figure.

Comments: Figure 6 shows a single mitochondrion with Drp-1 labeling. Even though this is a nice mitochondrion, it is not necessarily representative to the data described. The described results also do not indicate, whether there are significant changes of the Drp-1 labeling. This manuscript would benefit greatly if the micrographs in figure 2 and figure 6 could support each other.

Answer: We agree with your comment. The micrograph of Drp-1 labeling mitochondrion (Fig. 6) and the data on the number of DRP1-labeled mitochondria were obtained on a suspension of isolated mitochondria and therefore may be incorrect when compared with the main results of work. In this regard, we decided to remove this figure and its description from our manuscript. It does not diminish the value of the work. Indeed, our studies showed an increase in the expression level of Drp1 and a decrease in the expression of Mfn2 in diabetic animals as compared to control animals. These data are consistent with the results of morphometric analysis of mitochondria in liver tissue obtained using electron microscopy.

Comments: The authors use at times the term hepatocytes without any details of how these cells were identified as hepatocytes. The methods indicate the use of liver tissue for EM studies and to isolate mitochondria from. Since the liver contains 4 major liver cell types (hepatic stellate cells, hepatocytes, Kupffer cells, and liver sinusoidal endothelial cells), the authors should either re-word or add/reference how they obtained the hepatocytes.

Answer: Thanks a lot for the comment; the text has been corrected.

Minor: BG in table 2 is not defined. The paragraph beginning at line 72 contains a number of bullet points, but no 5 is missing.

Answer: All concerns have been corrected. Thank you for your time.